# Processing Effect and Characterization of Olive Oils from Spanish Wild Olive Trees (*Olea europaea* var. *sylvestris*)

**DOI:** 10.3390/molecules26051304

**Published:** 2021-02-28

**Authors:** Francisco Espínola, Alfonso M. Vidal, Juan M. Espínola, Manuel Moya

**Affiliations:** 1Department Chemical, Environmental and Materials Engineering, Universidad de Jaén, Paraje Las Lagunillas, Edif. B-3, 23071 Jaén, Spain; amvidal@ujaen.es (A.M.V.); mmoya@ujaen.es (M.M.); 2The University Hospital of Marqués de Valdecilla, Av. de Valdecilla, 25, 39008 Santander, Spain; juan1411994@hotmail.com

**Keywords:** Acebuchina, virgin olive oil, response surface methodology, volatile compounds, phenolic compounds

## Abstract

Wild olive trees have important potential, but, to date, the oil from wild olives has not been studied significantly, especially from an analytical point of view. In Spain, the wild olive tree is called “*Acebuche*” and its fruit “*Acebuchina*”. The objective of this work is to optimize the olive oil production process from the Acebuchina cultivar and characterize the oil, which could be marketed as healthy and functional food. A Box–Behnken experimental design with five central points was used, along with the Response Surface Methodology to obtain a mathematical experimental model. The oils from the Acebuchina cultivar meet the requirements for human consumption and have a good balance of fatty acids. In addition, the oils are rich in antioxidants and volatile compounds. The highest extraction yield, 12.0 g oil/100 g paste, was obtained at 90.0 min and the highest yield of phenolic compounds, 870.0 mg/kg, was achieved at 40.0 °C, and 90.0 min; but the maximum content of volatile compounds, 26.9 mg/kg, was obtained at 20 °C and 30.0 min. The oil yield is lower than that of commercial cultivars, but the contents of volatile and phenolic compounds is higher.

## 1. Introduction

The olive tree is an important economic crop in the Mediterranean basin and has remarkable cultural importance. The wild olive tree (*Olea europaea* var. *sylvestris*) is known as “*Acebuche*,” in Spain, and its fruit is known as “*Acebuchina*”. There is a close relationship between Acebuche and cultivated olive trees [1]. Acebuche are usually found in remote mountains and hard-to-reach places and have small oval fruits. Loureiro et al. [2] were the first to estimate the genome of six Portuguese varieties of olives, including the *sylvestris* cultivar. Several authors consider that the differences between cultivated and wild plants are minor [3,4], but the relationships between wild Mediterranean olives and cultivated olives remains unclear [5]. However, wild olive trees and cultivated olive trees can exchange genetic information.

Olive oil is highly appreciated for its flavor and high nutritional value. In addition, many people value its health benefits, and it is most useful edible oil in the world [6,7]. As part of a healthy diet, olive oil can have a protective effect against cardiovascular and inflammatory diseases [8,9].

The phenolic compounds contained in olive oil, along with other antioxidants, are responsible for the healthful properties attributed to these oils because they contribute to the protection of blood lipids from oxidative stress [10]. In addition, they are of special interest because they affect oil stability, taste, and aroma [11].

Among the phenolic compounds, oleacein and oleocanthal have drawn particular interest. These compounds are formed by the hydrolysis of oleuropein and ligstroside, respectively, during the elaboration of olive oil [12]. According to Beauchamp et al. [11], oleocanthal has anti-inflammatory properties, similar to drugs such as ibuprofen.

From a commercial point of view, the volatile compounds are also interesting because the taste and aroma of olive oils are dependent on these components. These compounds are also formed during the elaboration of olive oil, mostly through the action of enzymes that are released during the olive milling process.

Several factors influence the volatile compound content of olive oils, from agronomic and climatic factors to technological ones [13,14,15]. The three most important technological factors are the size of the hammer-mill sieve and the malaxation time and temperature. We can control these operating variables to obtain a high-quality oil that is rich in healthy phenolic compounds and has a high content of volatile compounds.

The main objective of this work is to characterize the olive oil obtained from the Acebuchina olives and compare it with the oil prepared from the main commercial variety of olives in the same area, the Picual olive. In addition, we would like to identify whether the chemical composition of Acebuchina oil is responsible for its differences with Picual oil. Therefore, the present work was performed to evaluate oil composition of Spanish wild olive oil.

In addition, we carried out a combined study of the three most important technological factors in the olive oil elaboration process to determine the optimal conditions to obtain the best quality oil and highest nutritional characteristics. In this respect, it is important to obtain highly aromatic olive oils having a good profile of volatile compounds and with a large phenolic compound content. For the optimization study, we used the statistical design of experiments and response surface methodology (RSM) [16].

## 2. Results and Discussion

The most relevant results and a detailed discussion of the same have presented in the next subchapters.

### 2.1. Extraction Yield, Quality Parameters, and Photosynthetic Pigments

Table 1 shows the extraction yield in grams of oil per 100 g of paste, and the model is shown in Table 2. Only the malaxation time had a statistically significant influence on the extraction yield. The maximum extraction yield was 12.0 g oil/100 g paste after 90.0 min, Table 3. This value is much lower than that of the commercial Picual cultivar; Espínola et al. [17] studied the Picual olive, which is the major commercial cultivar in the area of the study, and found an oil yield of 19.6% for a malaxation time of 89.5 min and for olives with a similar maturity index of 3.8.

Table 1 and Table 4 show some of the main parameters determined in the different analytical groups for all oils obtained using our experimental design. The acidity, K_232_, and K_270_ are not influenced by the operating factors. For the peroxide index, only the malaxation temperature and sieve size were influencing factors within the ranges studied for technological factors (Table 2). The peroxide index increases with temperature and decreases with increasing sieve size. All the values obtained from the quality parameters are within the limits established by EEC [18]. Therefore, the oil obtained from Acebuchina is suitable for human consumption.

In the models for the chlorophyll and carotenoid contents (Table 2), all technological factors had an influence. The response surface for the chlorophyll content is shown in Figure 1 for a sieve size of 5.5 mm. As shown in the figure, the chlorophyll content increases with both time and temperature, but the temperature has a stronger influence. On the other hand, the chlorophyll content decreases with increase in sieve size. The maximum values found for chlorophylls and carotenoids were 51.5 and 24.8 mg/kg, respectively (Table 3). Thus, these responses depend very much on the ripeness index of the olives, but we observed similar values with the oil from the Picual cultivar [17], and the values are higher than those observed by Hannachi et al. [19] for oils obtained from Tunisian wild olives.

### 2.2. Fatty Acids

The total content of FAs in the oils was determined, and the results are listed in Table 4. The sums of monounsaturated fatty acids (MUFA), polyunsaturated fatty acids (PUFA), saturated fatty acids (SFA), as well as the C18:1/C18:2 and MUFA/PUFA ratios, were also determined, but no statistically significant mathematical model was obtained. Therefore, we conclude that, in the ranges studied, none of the three technological factors influences the content or composition of fatty acids in oils. Table 5 shows the mean values obtained for all oil samples and a comparison with the ranges established by the IOC and the EU for virgin olive oils: the results obtained are within the ranges established by both organizations. The oleic acid content obtained from Acebuchina oil, 76.90%, is higher than those obtained by Hannachi et al. [19] for oils obtained from Tunisian wild olives and [20] for oils obtained from Pakistani wild olives. On the other hand, Bouarroudj et al. [21] confirmed that there is a great variation between the parameters measured for olive oils from different wild olive trees, including the fatty acid composition.

### 2.3. Phenolic Compounds and Antioxidant Activity

Phenolic compounds, together with tocopherols, are responsible for the antioxidant capacity of olive oil. Phenolic compounds are also associated with its bitter taste [22]. Table 4 lists the contents of the main group of phenolic compounds, the secoiridoids, and the total phenols as a sum of the individual phenolic compounds. In addition, Table 4 shows the antioxidant potential (determined using the DPPH assay). Table 2 and Table 6 show the mathematical models obtained. According to the models, among the technological factors studied, the malaxation temperature had the greatest influence, and the antioxidant potential generally increased with increase temperature (no influence was observed for hydroxytyrosol, tyrosol, and *p*-coumaric acid). Some phenolic compounds were also affected to a lesser extent by the sieve size and the malaxation time. Figure 2 shows the total phenol high-performance liquid chromatography (HPLC) model versus malaxation time and temperature. This figure shows an increase in phenol content with temperature, and this was more pronounced at longer times. Likewise, for the variation with the malaxation time, a negative influence at low temperatures and a positive influence at high temperatures is observed. This change in action on the response depending on temperature is due to the strong interaction between the factors. Our results are consistent with those reported by other authors [23,24]. However, according to Ben Brahim et al. [25], the phenol contents is not significantly influenced by the malaxation time. These differences can be explained by the interaction between the temperature and the malaxation time.

The maximum content of phenolic compounds predicted by the model is 870.0 mg/kg (Table 3) at 40.0 °C and 90 min of malaxation. Comparing these values with those of oils obtained using commercial varieties of olives, it can be seen that the content of phenolic compounds in Acebuchina oil is very high and slightly higher than that found for Picual oils, 813.5 mg/kg [26].

The secoiridoid derivatives (3,4-DHPEA-EDA, *p*-HPEA-EDA, 3,4-DHPEA-EA, and *p*-HPEA-EA) were the major phenolic compounds and increased with increase in temperature. This is a similar result to those determined by other authors [27,28,29]. Figure 3 shows the response surface for oleocanthal (*p*-HPEA-EDA) content versus malaxation time and temperature. On the contrary, some authors reported that the maximum phenolic content is obtained at the shorter possible malaxation time [30].

The antioxidant potential is shown in Table 4. According to the model listed in Table 2, the antioxidant potential is correlated positively to the size of hammer-mill sieve and malaxation temperature but not to the malaxation time. The fact that we obtained different models for phenolic compounds and antioxidant potential may be because, in oils, there are a range of other antioxidant compounds than phenols. Our results are consistent with those obtained by Bouarroudj et al. [21] for oil from Algerian wild olives.

### 2.4. Volatile Compounds

Table 4 shows the total content of volatile compounds formed via the lipoxygenase (LOX) pathway, as well as those of some individual volatile compounds. Furthermore, an example of a chromatogram is shown in Appendix A. Table 2 and Table 7 show the models for volatile compounds, and Figure 4 shows the response surface for total LOX volatiles content at a malaxation time of 60 min. According to the model in Table 2 and response surface in Figure 4 for total LOX volatile content, the malaxation temperature is the factor that most influences (in a negative fashion) the response. Time negatively influences the total LOX content, whereas the sieve size positively affects it, although the influence is slight in both cases. In these oils, *cis*-3-hexenyl acetate is present in higher quantities than other commercial oils. In the models obtained for individual volatile compounds of the LOX pathway, there is a decrease in LOX compounds with increase in temperature, and there is also a slight increase as the size of the sieve increases. Our results are in agreement with those of several authors [29,31,32] and may result from the inactivation of hydroperoxide lyase enzymes [33,34,35]. We have also observed this with other olive cultivars [36].

The maximum amount of volatile compounds determined by the model is 26.9 mg/kg (Table 3) using the lowest malaxation temperature and shortest time (20.0 °C and 30 min) and the largest sieve size (6.5 mm). In comparison with the values obtained for commercial varieties of olives, the content of volatile compounds in the Acebuchina oils is very high, practically double that of Picual oils, at 12.7 mg/kg [26].

## 3. Conclusions

The olive oil obtained from Acebuchina is suitable for human consumption and all the quality parameters comply with current European Union (EU) and international regulations (IOC). Qualitatively, Acebuchina and Picual oils have an identical composition. Therefore, wild olives could be an interesting source of edible oil.

In this work, we have also determined the content and composition of fatty acids, phenolic and volatile compounds, and photosynthetic pigments. We have confirmed that the oil is rich in volatile and phenolic compounds. Therefore, Acebuchina oil is commercially promising because of its high antioxidant potential and could be marketed from a nutritional, medicinal, and even cosmetic point of view.

In the studied temperature range (20–40 °C), the technological factor that most influences the phenolic compounds and antioxidant activity is the malaxation temperature, and we observed that the phenolic compounds and the antioxidant activity increased when the temperature was increased. In contrast, the volatile compound content was reduced with increase in malaxation temperature.

The results of this study suggest that the content of volatile compounds in the Acebuchina oils are very interesting, practically double that the values obtained for commercial cultivars of olives grown in the area. The malaxation temperature was the factor that had the greatest influence on the total LOX pathway volatile compounds. Lower malaxation temperature resulted in a higher quantity of volatile compounds.

## 4. Materials and Methods

### 4.1. Plant Material and Oil Extraction

The Acebuchina olives (*Olea europaea* subsp. *europaea* var. *sylvestris*) were collected in Puente de la Sierra (Jaen, Southern Spain). The wild olives were sampled in the wild terrain of the mountain, although near the site of Picual olives trees. The olives were hand-picked in November 2018. Then, fruits with a maturity index of 3.5 were immediately transported to the laboratory. The fruit maturity index was determined following the method described by Espínola et al. [37].

Following the protocol defined by the European Economic Community [18], the oil content was determined to be 14.7 g oil/100 g of paste using the Soxhlet method. The extraction duration with hexane was 6 h and the sample amount to be extracted was approximately 10 g. The moisture was found to be 55.4 g/kg after drying the olive paste at 105 °C.

Olives were processed under laboratory-scale conditions using an Abencor centrifugal system described by Espínola et al. [17]. The extracted oils were decanted and filtered with paper; later, the oils were stored in amber glass bottles, under N_2_ atmosphere at −18 °C until analysis.

### 4.2. Experimental Design

A Box-Behnken design with five central points for three factors (the size of hammer-mill sieve and malaxation time and temperature) was used (see Table 1) to determine the influences of these technological factors on fifty-four different responses obtained from the olive oils. The malaxation time was varied from 30 to 90 min, the temperature was varied from 20 to 40 °C, and the hammer-mill sieve size was varied from 4.5 to 6.5 mm. A quadratic model was used for each response according to Equation (1) using Design-Expert ver. 8.0.7.1 (Stat-Ease, Inc., Minneapolis, MN, USA) software:*Y = β*_0_*+ β*_1_ D *+ β*_2_ T *+ β*_3_ t *+ β*_12_ D T *+ β*_13_ D t *+ β*_23_ T t *+ β*_11_ D^2^*+ β*_22_ T^2^*+ β*_33_ t^2^ ± SD(1)
where, *Y* is the response, D is the size of hammer-mill sieve (sieve hole diameter, mm), T is the malaxation temperature (°C), t is the malaxation time (min), and SD is the standard deviation, considered as the model error.

### 4.3. Statistical Analysis

Statistical parameters were determined using the analysis of variance (ANOVA). The statistical significance of the models and their coefficients were evaluated at the 5% probability level (*p*-value < 0.05). All models were statistically significant, and there was no lack of fit (lack of fit > 0.05).

### 4.4. Analytical Methods

The main quality parameters were determined according to EEC [18] (peroxide index, acidity, and spectrophotometric indexes K_232_ and K_270_).

#### 4.4.1. Determination of Chlorophyll and Carotenoid

To determine the chlorophyll and carotenoid content, the procedure proposed by Mínguez-Mosquera [38] was used together with Equations (2) and (3). The results are expressed in milligrams of photosynthetic pigment per kilogram of oil.
(2)$$\mathrm{Chlorophyll}\;=\left(A_{670}\;\times \;10^{6}\right)/\left(E_{0}\;\times \;100\;\times \;d\right)$$
(3)$$\mathrm{Carotenoid}\;=\left(A_{470}\;\times \;10^{6}\right)/\left(E_{0}\;\times \;100\;\times \;d\right)$$
where, *A* is the absorbance, *E*_0_ = 613 for chlorophyll, *E*_0_ = 2000 for carotenoid, and *d* is the spectrophotometer cell thickness (1 cm). The spectrophotometer used to measure the absorbance was a UV-1800 spectrophotometer (Shimadzu, Kyoto, Japan).

#### 4.4.2. Determination of Fatty Acid

The fatty acid (FA) content was determined according to EEC [18], and the details of the analytical method have been described by Vidal et al. [39]. The chromatographic separation was performed using a model 7890B-GC gas chromatograph (Agilent Technologies, Santa Clara, CA, USA) and a capillary column HP-88 (60 m length; 0.25 mm internal diameter; 0.2 mm coating; Agilent Technologies). The injector and detector (FID) temperature was 250 and 260 °C, respectively. Helium was used as carrier gas. 1 μL of sample was injected in split mode (1:100). The initial temperature of oven was 100 °C during 5 min. A ramp of 4 °C/min was used to increase the temperature until 240 °C and this was kept during 30 min. A standard mixture of FAs (Supelco, Bellefonte, PA, USA) was used to identify the FAs in the oils.

#### 4.4.3. Determination of Volatile Compounds

The method used to determine volatile compounds in the oils has been described by Vidal et al. [26]. The SPME fiber was composed of Carboxen/DVB/polydimethylsiloxane, supplied by Supelco and its characteristics were 2 cm length and 50/30 μm film thickness. A 7890B-GC gas chromatograph (Agilent Technologies) was used to perform the analysis. A DB-WAXetr capillary column was used (Agilent Technologies, 30 m length, 0.25 mm internal diameter, 0.25 μm coating). The flow rate was 1 mL/min of helium as carrier gas. The injector and detector (FID) temperatures were 260 and 280 °C, respectively. Initially, the oven temperature was held at 40 °C during 10 min. A ramp of 3 °C/min was then used to increase the temperature until 160 °C. Furthermore, another ramp of 15 °C/min was used to increase the temperature from 160 °C to 200 °C and this was maintained for 5 min. Two grams of sample were analyzed by headspace–solid phase microextraction (HS-SPME). 4-Methyl-2-pentanol was used as the internal standard, and 39 external standards (listed in the Supplementary Material) were used. The results are expressed in milligrams of each compound standard per kilogram of oil.

#### 4.4.4. Determination of Phenol Compounds

To identify phenol compounds, the method proposed by the International Olive Council [40] was used. Details of the analytical method used has been described in [26]. A HPLC system (Shimadzu, Kyoto, Japan) equipped with a BDS Hypersil C18 column (particle size 5 μm, 25 cm length, and 4.6 mm internal diameter; Thermo Scientific, Waltham, Massachusetts, USA) was used. The mobile phase was water with 0.2% of orthophosphoric acid (A), methanol (B), and acetonitrile (C). A flow ramp was used to change the phases proportions. Initially, phase A proportion was 96%, B and C were 2%. At minute 40, phase A proportion was 50%, B and C were 25%. At minute 45, phase A proportion was 40%, B and C were 30%. At minute 60, phase A proportion was 0%, B and C were 50%. From minute 72 to 80, the phases proportions were again as the initial ones. The elution flow was 1 mL/min. The oven temperature was 30 °C and 20 μL of sample was injected. The signal was registered by the detector UV at 280 nm. Results are expressed in milligrams of tyrosol per kilogram of oil.

#### 4.4.5. Antioxidant Potential

To determine the antioxidant potential, the 2,2-diphenyl-1-picrylhydrazyl (DPPH) free radical scavenging assay was used. This has been previously described in [26]. The percentage of DPPH radicals scavenged was calculated according to Equation (4):(4)$$\%DPPH_{inhibition}=\frac{{\left[DPPH\right]}_{0}-{\left[DPPH\right]}_{sample}}{{\left[DPPH\right]}_{0}}\times 100$$
where, *[DPPH]*_0_ is control concentration, and *[DPPH]_sample_* is sample concentration The percentage inhibition was converted to the antioxidant activity using Trolox as the standard antioxidant.

## Figures and Tables

**Figure 1 molecules-26-01304-f001:**
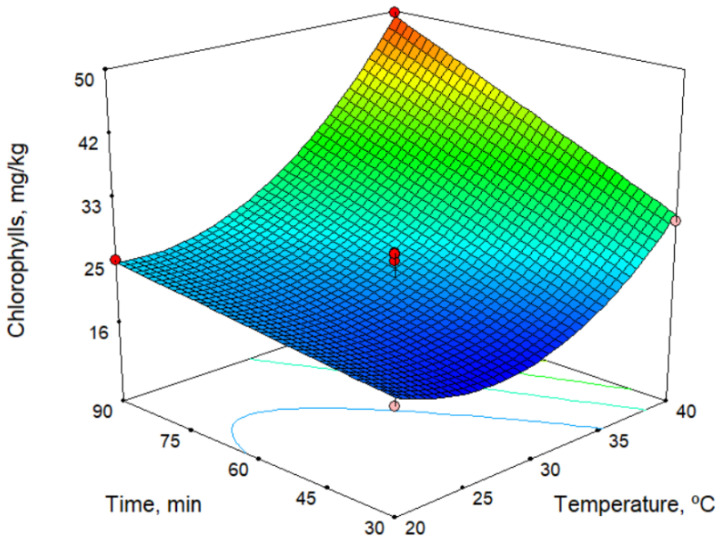
Response surface for chlorophylls content, for sieve size 5.5 mm.

**Figure 2 molecules-26-01304-f002:**
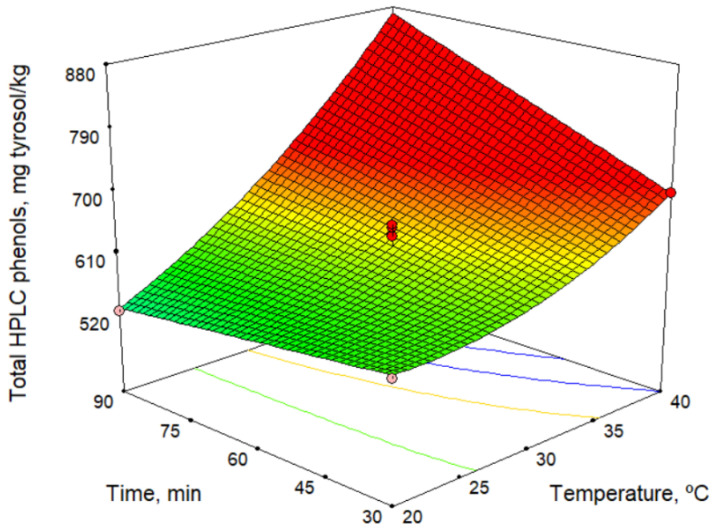
Response surface for total HPLC phenols content.

**Figure 3 molecules-26-01304-f003:**
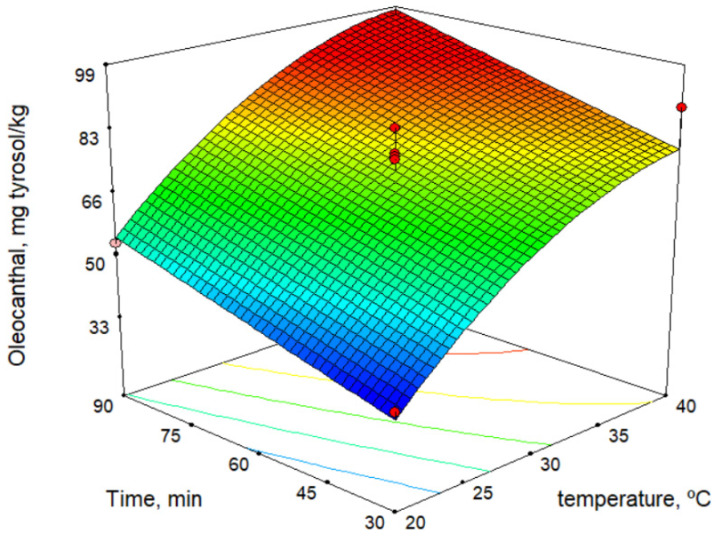
Response surface for oleocanthal content.

**Figure 4 molecules-26-01304-f004:**
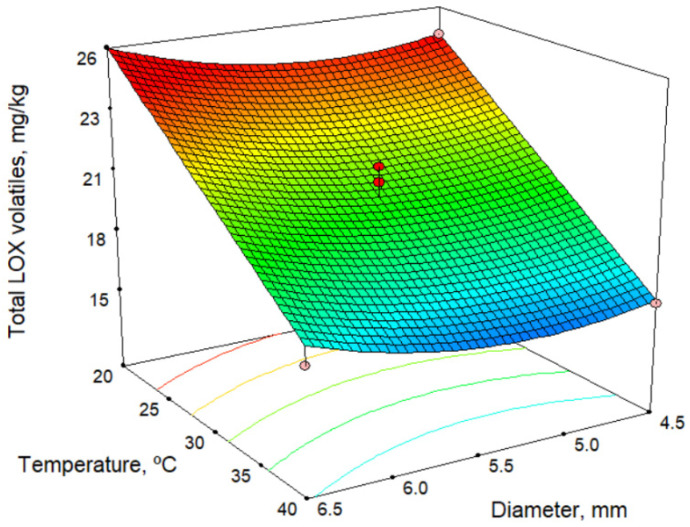
Response surface for total LOX volatiles content, for malaxation time 60 min.

**Table 1 molecules-26-01304-t001:** Experimental design and responses for Acebuchina virgin olive oil.

	Actual Factors			Responses		
Design Points	Diameter (mm)	Temperature (°C)	Time (min)	Extraction Yield (g oil/100 g paste)	Acidity (%)	Peroxide Index (mEq O_2_/kg)
1	5.5	30	60	11.9	0.19	0.17
2	5.5	20	90	12.1	0.23	0.22
3	4.5	30	90	11.5	0.25	0.23
4	6.5	30	90	10.7	0.20	0.16
5	6.5	40	60	12.1	0.18	0.31
6	5.5	20	30	10.6	0.19	0.17
7	6.5	20	60	11.6	0.21	0.18
8	5.5	40	90	11.9	0.20	0.25
9	4.5	40	60	12.0	0.21	0.30
10	5.5	40	30	10.6	0.23	0.28
11	6.5	30	30	11.1	0.19	0.17
12	4.5	30	30	11.0	0.22	0.24
13	5.5	30	60	11.7	0.22	0.17
14	4.5	20	60	11.4	0.22	0.23
15	5.5	30	60	11.4	0.22	0.23
16	5.5	30	60	11.6	0.23	0.19
17	5.5	30	60	11.8	0.19	0.26

**Table 2 molecules-26-01304-t002:** Models (Equation (1)) in terms of actual factors and statistical parameters for the responses in Table 1 and Table 4. Analysis of Variance (ANOVA) for the fit of experimental data was used.

Response	Model *	*p*-Value	R^2^	SD
Extraction Yield (g oil/100 g paste)	9.3471 + 0.05929 t − 3.2956 × 10^−4^ t^2^	<0.0001	0.802	0.24
Acidity (%)	0.2102	-	-	0.019
Peroxide index (mEq O_2_/kg)	0.55609 − 0.030919 D − 0.015337 T + 3.0769 × 10^−4^ T^2^	0.0016	0.707	0.026
K_232_	1.2552	-	-	0.13
K_270_	0.15327	-	-	0.29
Chlorophylls (mg/kg)	87.371 − 2.2140 D − 4.4013 T − 0.16206 t + 0.011715 T t + 0.076590 T^2^	<0.0001	0.964	2.07
Carotenoids (mg/kg)	35.666 − 1.10962 D − 1.2610 T − 0.011458 t + 0.0028785 T t + 0.022051 T^2^	<0.0001	0.929	0.91
Total LOX volatiles pathway (mg/kg)	63.072 − 11.067 D − 0.41160 T − 0.03092 t + 1.0636 D^2^	< 0.0001	0.931	0.84
Total HPLC phenols (mg tyrosol/kg)	869.94 − 20.080 T − 4.5593 t + 0.18558 T t + 0.34121 T^2^	0.0005	0.926	17.22
DPPH (µmol Trolox/kg)	54.733 + 308.72 D + 44.108 T	0.0012	0.815	149.2

* According to coefficients of model (Equation (1)), only significant values are included (*p*-value < 0.005). D is the size of hammer-mill sieve (mm). T is the malaxation temperature (°C). t is the malaxation time (min). R^2^ is the coefficient of determination. SD is the standard deviation.

**Table 3 molecules-26-01304-t003:** Optimal conditions for the maximum of the main responses.

Individual Response	Maximum Value	Diameter (mm)	Temperature (°C)	Time (min)
Extraction Yield (g oil/100 g paste)	12.0	4.5–6.5	20–40	90.0
Chlorophylls (mg/kg)	51.5	4.5	40.0	90.0
Carotenoids (mg/kg)	24.8	4.5	40.0	90.0
Total LOX volatiles (mg/kg)	26.9	6.5	20.0	30.0
*trans*-2-Hexenal (mg/kg)	7.10	4.5–6.5	28.2	30.0
*cis*-3-Hexenyl acetate (mg/kg)	8.30	4.5–6.5	20–40	30–90
Total HPLC phenols (mg tyrosol/kg)	870.0	4.5–6.5	40.0	90.0
Oleacein (mg tyrosol/kg)	379.4	4.5–6.5	40.0	30–90
Oleocanthal (mg tyrosol/kg)	98.3	4.5–6.5	40.0	90.0
DPPH (µmol Trolox/kg)	3826	6.5	40.0	30–90

**Table 4 molecules-26-01304-t004:** Selected responses, from the different analysis groups carried out, of the oils obtained in the experimental design

Design Points	1	2	3	4	5	6	7	8	9	10	11	12	13	14	15	16	17
**Fatty Acid (g/100 g oil)**																	
Palmitic acid (C16:0)	13.48	13.54	13.49	13.51	13.51	13.55	13.54	13.69	13.54	13.58	13.57	13.60	13.47	13.47	13.48	13.48	13.48
Stearic acid (C18:0)	2.41	2.39	2.40	2.44	2.39	2.38	2.40	2.44	2.44	2.40	2.40	2.41	2.42	2.38	2.42	2.42	2.42
Oleic acid (C18:1)	76.90	77.64	76.56	78.07	77.32	77.17	76.56	75.42	76.65	76.41	77.04	76.37	76.43	76.77	77.68	77.50	77.64
Linoleic acid (C18:2)	4.66	3.80	4.93	3.39	4.14	4.24	4.88	5.79	4.75	5.01	4.34	4.99	5.10	4.75	3.84	4.03	3.89
Linolenic acid (C18:3)	0.36	0.35	0.35	0.35	0.36	0.35	0.35	0.35	0.36	0.36	0.35	0.35	0.36	0.36	0.36	0.36	0.37
**Phenolic Compounds** **(mg tyrosol/kg)**																	
3.4-DHPEA-EDA (oleacein)	338.90	263.36	317.49	354.34	188.82	320.59	313.96	241.45	212.39	386.11	342.11	341.43	345.37	249.00	344.53	308.48	334.47
3.4-DHPEA-EA	92.35	76.55	97.47	102.37	65.60	72.33	71.21	70.67	65.78	100.78	77.17	75.93	94.47	68.83	89.60	87.88	89.75
p-HPEA-EDA (oleocanthal)	75.74	53.00	87.73	90.40	87.72	35.82	41.51	88.97	86.78	88.21	57.44	55.65	74.96	46.71	70.00	76.45	83.10
p-HPEA-EA	15.89	12.93	18.48	17.44	20.25	9.77	10.47	19.17	17.05	16.86	10.96	11.18	16.07	11.50	14.97	15.91	17.29
**Total HPLC phenols**	635.38	523.99	662.65	671.65	429.05	574.86	563.05	487.65	453.76	698.61	607.28	606.04	652.14	484.93	648.53	597.34	636.13
**DPPH (µmol trolox/kg)**	3190	2605	2745	3441	2272	3070	3023	2268	1959	3566	3250	2790	2653	2668	3322	2870	2902
**Volatile Compounds (mg/kg)**																	
hexanal	1.04	1.36	1.23	0.98	1.01	1.08	0.99	1.02	1.10	1.23	0.94	1.02	0.95	1.47	0.93	0.97	1.09
*trans*-2-hexenal	6.71	6.93	6.11	6.46	3.82	6.15	6.06	3.77	4.56	5.56	7.11	6.88	6.58	7.17	7.18	6.66	6.70
*cis*-3-hexenyl acetate	9.20	9.57	7.97	9.41	7.51	9.02	8.39	6.32	6.79	7.14	10.18	8.62	6.82	11.18	7.31	7.40	8.22
**Total LOX volatiles pathway**	21.50	23.11	20.04	21.72	16.94	21.69	20.77	14.97	16.33	17.84	22.84	21.36	18.78	24.69	19.87	19.46	20.82
acetic acid	1.20	1.15	1.31	1.17	1.17	1.28	1.32	1.31	1.11	0.97	0.76	0.83	0.80	0.89	0.95	1.10	1.09
Nonanal	3.23	3.44	3.40	3.42	3.53	2.93	2.78	3.14	2.99	2.71	2.53	2.60	2.80	3.05	2.70	2.80	3.30

**Table 5 molecules-26-01304-t005:** Fatty acid composition of Acebuchina olive oils (% *w*/*w* methyl ester) *.

Fatty Acids	Acebuchina	IOC	EU
Myristic acid	--	<0.03	<0.03
Palmitic acid (C16:0)	13.50 ± 0.06	7.50–20.00	-
Palmitoleic acid (C16:1)	1.65 ± 0.04	0.30–3.50	-
Heptadecanoic acid	0.05 ± 0.00	<0.40	-
Heptadecenoic acid	0.11 ± 0.00	<0.60	-
Stearic acid	2.41 ± 0.02	0.50–5.00	-
Oleic acid	76.90 ± 0.66	55.00–83.00	-
Linoleic acid	4.50 ± 0.61	2.50–21.00	-
Linolenic acid	0.36 ± 0.01	<1.00	<1.00
Arachidic acid	0.05 ± 0.01	<0.60	<0.60
Gadoleic acid (eicosenoic)	0.24 ± 0.01	<0.50	<0.50
Behenic acid	0.11 ± 0.00	<0.20	<0.20
Lignoceric acid	0.05 ± 0.00	<0.20	<0.20
MUFA	78.95 ± 0.65		
PUFA	4.86 ± 0.61		
SFA	16.19 ± 0.07		
C18:1/C18:2	17.42 ± 2.57		
MUFA/PUFA	16.51 ± 2.26		

* Values are means ± standard deviations. IOC: Reference value of extra virgin olive oil formulated by International olive council (COI/T.15/NC No 3/Rev. 11 of July 2016). EU: Reference value of extra virgin olive oil formulated by REGULATION (EU) 2016/2095 of 26 September 2016. MUFA: monounsaturated fatty acid. PUFA: polyunsaturated fatty acid. SFA: saturated fatty acid. C18:1/C18:2 ratio. MUFA/PUFA ratio.

**Table 6 molecules-26-01304-t006:** Models (Equation (1)) in terms of actual factors and statistical parameters for the responses in Table 4. Analysis of Variance (ANOVA) for the fit of experimental data was used.

Response	Model *	*p*-Value	R^2^	SD
**Phenol compounds (mg/kg)**				
hydroxytyrosol	1.55435	--	--	0.46
Tyrosol	2.9336	--	--	0.26
Vainillin	−0.57634 + 0.024602 T + 0.012197 t	0.0003	0.768	0.17
*p*-coumaric acid	1.33283	--	--	1.58
*trans*-ferulic acid	−2.9983 + 0.24869 T	<0.0001	0.887	0.61
3.4-DHPEA-EDA (oleacein)	214.72 + 4.1176 T	0.0022	0.626	19.3
3.4-DHPEA-EA	−154.096 + 14.155 T + 0.85230 t − 0.23996 T^2^ − 4.01759 × 10^−3^ t^2^	<0.0001	0.976	2.26
*p*-HPEA-EDA (oleocanthal)	−88.2736 + 7.29244 T + 0.34579 t − 0.085155 T^2^	<0.0001	0.913	6.06
*p*-HPEA-EA	−0.48650 + 0.35825 T + 0.080159 t	<0.0001	0.857	1.33
pinoresinol	17.973 + 0.36258 D + 0.7828 T − 0.16069 t − 0.01859 T^2^ + 8.658 × 10^−4^ t^2^	<0.0001	0.989	0.37
luteolin	−3.52249 + 0.54037 T − 0.010703 T^2^	<0.0001	0.943	0.22
apigenin	4.35334 + 0.41585 T − 8.61064 × 10^−3^ t − 8.54698 × 10^−3^ T^2^	<0.0001	0.892	0.31

* According to coefficients of model (Equation (1)), only significant values are included (*p*-value < 0.005). D is the size of hammer-mill sieve (mm). T is the malaxation temperature (°C). t is the malaxation time (min). R^2^ is the coefficient of determination. SD is the standard deviation

**Table 7 molecules-26-01304-t007:** Models (Equation (1)) in terms of actual factors and statistical parameters for the responses in Table 4. Analysis of Variance (ANOVA) for the fit of experimental data was used.

Response	Model *	*p*-Value	R^2^	SD
**Volatile compounds (mg/kg)**				
LOX pathway				
hexanal	3.8430 − 0.39935 D − 0.11835 T + 0.013683 t + 9.64688 ×·10^−3^ D T − 4.12082 ×·10^−4^ T t + 1.38944 × 10^−3^ T^2^	0.0004	0.881	0.066
hexan-1-ol	0.49756	--	--	0.042
*trans*-2-hexenal	−4.18051 + 0.74575 T + 0.054078 t − 2.1411199 ×·10^−3^ T t − 0.012079 T^2^	<0.0001	0.924	0.35
*trans*-2-hexen-1-ol	1.08961 + 0.035689 D − 0.012801 T − 1.73427 ×·10^−3^ t	<0.0001	0.810	0.054
*cis*-3-hexen-1-ol	7.04195 − 2.13938 D − 0.013436 t + 2.54523 ×·10^−3^ D t + 0.18685 D^2^	0.0003	0.804	0.065
*cis*-3-hexenyl acetate	8.29684	--	--	1.33
1-penten-3-ol	1.2222 − 0.021669 T	<0.0001	0.965	0.029
1-penten-3-one	1.3926 − 0.01641 T	<0.0001	0.831	0.052
*cis*-2-penten-1-ol	0.96444 − 8.03020·× 10^−3^ T + 1.44138 ×·10^−3^ t	0.0004	0.670	0.048
*trans*-2-pentenal	0.47230	--	--	0.038
Sugar fermentation				
acetic acid	1.08262	--	--	0.18
Other compounds				
pentan-3-one	0.3738	--	--	0.041
octanal	0.64314	--	--	0.057
nonanal	3.01463	--	--	0.32

* According to coefficients of model (Equation (1)), only significant values are included (*p*-value < 0.005). D is the size of hammer-mill sieve (mm). T is the malaxation temperature (°C). t is the malaxation time (min). R^2^ is the coefficient of determination. SD is the standard deviation.

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
