# Peer review of "Processing Effect and Characterization of Olive Oils from Spanish Wild Olive Trees (*Olea europaea* var. *sylvestris*)"

_molecules, 2021, doi:10.3390/molecules26051304_

Round 1
Reviewer 1 Report
The article titled Elaboration and characterization of olive oils from Spanish wild olive trees (Olea europaea var. sylvestris) seems to be very interesting. The article is well written, the introduction contains important data taking into account the latest knowledge. The results are very well presented. The discussion is clearly described. However, below I am sending a few considerations:
- There is no summary in the abstract
- The description of the soxhlet total fat determination should be better described. Extraction conditions are not given.
- Statistical analysis should be a separate subchapter
- Analytical methods should be separated, more details should be provided.
- There should be an introductory sentence at the beginning of the discussion.
- line 228 is underlined in degrees Celsius. Should be removed.
- In tables, use left justification for the first column.
Author Response
Dear Reviewer 1, our answers in attached file.
Best regards.

Reviewer 2 Report
Comments:
1, Any other volatile compounds other than the 39 compounds with standards was detected in the oils, such as acetone or other common volatile organics? If yes, what are the compounds? Comparing to commercial olive oil, those compounds are higher or lower on concentration? If no, why authors don’t say so?
2, Line 112-113, “… 39 external standards were used.”
What are the 39 compounds? What kind of detected limits for the volatile compounds? Table 4 only lists very few volatile compounds.
3, Authors didn’t provide a typical GC plot for the samples. It is better to give an example in supplemental materials. The information in comment #2 can be provided in supplemental materials too.
4, Line 182-183, “Likewise, for the variation with the malaxation time, a negative influence at low temperatures and a positive influence at high temperatures is observed.”
In figure 2, it is very hard to see “a negative influence at low temperatures”. Is this a non-significant effect or a mistake?
5, Tables
For the constant numbers in tables (model), some of them have two significant digits after the point. Some have five significant digits after the point. What are the reasons behind that? Should all of the constant numbers in each of the models have two significant digits after the points?
Author Response
Dear Reviewer 2, our answers in attached file.
Best regards.
